# Improper electric polarization in simple perovskite oxides with two magnetic sublattices

Hong Jian Zhao[1,2,3], L. Bellaiche[3], Xiang Ming Chen[2] & Jorge Íñiguez[1,4]

$ABO_3$ perovskite oxides with magnetic $A$ and $B$ cations offer a unique playground to explore interactions involving two spin sublattices and the emergent effects they may drive. Of particular interest is the possibility of having magnetically driven improper ferroelectricity, as in the much studied families of rare-earth orthoferrites and orthochromites; yet, the mechanisms behind such effects remain to be understood in detail. Here we show that the strongest polar order corresponds to collinear spin configurations and is driven by non-relativistic exchange-strictive mechanisms. Our first-principles simulations reveal the dominant magnetostructural couplings underlying the observed ferroelectricity, including a striking magnetically driven piezoelectric effect. Further, we derive phenomenological and atomistic theories that describe such couplings in a generic perovskite lattice. This allows us to predict how the observed effects can be enhanced, and even how similar ones can be obtained in other perovskite families.

[1] Materials Research and Technology Department, Luxembourg Institute of Science and Technology (LIST), 5 avenue des Hauts-Fourneaux, L-4362 Esch/Alzette, Luxembourg. [2] Laboratory of Dielectric Materials, School of Materials Science and Engineering, Zhejiang University, Hangzhou 310027, China. [3] Physics Department and Institute for Nanoscience and Engineering, University of Arkansas, Fayetteville, Arkansas 72701, USA. [4] Institut de Ciència de Materials de Barcelona (ICMAB-CSIC), Campus UAB, 08193 Bellaterra, Spain. Correspondence and requests for materials should be addressed to H.J.Z. (email: hongjian.zhao@list.lu) or to L.B. (email: laurent@uark.edu) or to J.Í. (email: jorge.iniguez@list.lu).

The $RFeO_3$ and $RCrO_3$ compounds, where $R$ is a magnetic lanthanide, are usually called orthoferrites and orthochromites, respectively, as they all present the orthorhombic (*Pbnm*, GdFeO$_3$-type) structure[1,2] that is most common among perovskite oxides. These materials have received continued attention over decades because of their intriguing magnetic properties (see, for example, refs 3–16 and references therein). More recently, they have regained interest in connection with prospect applications in magnetic devices, and because some of them have been shown to present ferroelectric order at low temperatures, which renders them multiferroic[6–9,15]. More generally, the $RFeO_3$ and $RCrO_3$ materials constitute a unique playground to explore the novel physical effects potentially emerging from the presence of two interacting spin sublattices, and—thanks to their insulating character—the possibility of tuning such effects by the application of external electric fields.

In spite of the intense activity focused on these compounds, some of their key properties remain poorly understood at a fundamental, atomistic level. For example, in what regards their magnetism, a thorough characterization has been available for a long time[3,4]; yet, the key role that certain magnetostructural couplings play in determining the ground state structure and other remarkable effects—as, for example, the presence of a magnetic compensation temperature at which the magnetizations of the rare-earth ions and Fe/Cr sublattice cancel each other—has been discussed and highlighted only recently[16]. Similarly, the onset of a ferroelectric polarization at low temperatures has so far been addressed mainly at an empirical level, and conflicting pictures (for example, as regards the alleged need of a non-zero canted magnetization in order to have ferroelectricity[11,13]) can be found in the literature. Further, to the best of our knowledge only one first-principles work has discussed this problem[10], and that study was done under some simplifying assumptions (that is, adopting an A-type anti-ferromagnetic order for the Fe spins of DyFeO$_3$, while it is known that the actual arrangement is dominantly G-type). Hence, while valuable, the existing works and discussions are hardly complete or conclusive, and there is a clear need for a systematic investigation of the origin of ferroelectricity in orthoferrites and orthochromites.

In this work we use first-principles methods to address this pending issue, adopting a general approach whose conclusions should, in fact, be relevant to all orthoferrites and orthochromites, and even all $ABO_3$ perovskites with $A$ and $B$ magnetic sublattices. Our study shows there are two types of mechanisms leading to a ferroelectric polarization in these materials: a first type for which the effect is small and emerges from the symmetry breaking associated to a non-collinear spin structure; and a second type that leads to much larger polarizations and relies on exchange-strictive effects displayed by strictly collinear magnetic configurations. Here we mostly focus on the latter strongest effect and analyse the magnetic couplings responsible for the electric polarization, showing that they are mediated by up to four different structural distortions characteristic of all orthorhombic *Pbnm* perovskite phases. Most interestingly, we reveal a magnetically driven piezoelectric effect as one of the dominant sources of polarization. We discuss the atomistic models explaining the most interesting effects, and explain the implications of our results for other perovskite families and the engineering of new or optimized multiferroics.

## Results

**Conditions for ferroelectricity.** Existing evidence clearly shows that ferroelectricity in orthoferrites and orthochromites is driven by the onset of magnetic order in both the rare-earth and transition-metal sublattices[6,15]. This is thus a case of magnetically

induced ferroelectricity, via symmetry breaking, which places these materials among the so-called Type II multiferroics[17]. To understand the details of these symmetry breakings, our first task is to determine under which specific conditions—that is, for which particular spin arrangements—the electric polarization appears.

To do that we consider all the spin configurations that are potentially relevant in these materials, following experimental evidence that: first, the dominant order in the Fe/Cr sublattice is anti-ferromagnetic G-type (G-AFM), with first-nearest-neighboring spins anti-parallel to each other, and second, the typical spin structures are commensurate with the 20-atom crystallographic cell deduced from the atomic positions. More specifically, for the $R$ spins we consider G-AFM, C-AFM, A-AFM and ferromagnetic (FM) configurations (see the respective sketches in Fig. 1c). Note that for the C-AFM and A-AFM cases we consider the particular spin arrangement that is compatible with the crystallographic cell; this implies that the $c$ axis of in-phase rotations coincides with the so-called unique axis of these spin structures (that is, the direction of parallel spin alignment for the C-AFM case, and anti-parallel alignment for the A-AFM order). We consider three possible orientations (along the orthorhombic axes indicated in Fig. 1b) of the vector describing the dominant AFM order, and allow the $R$ and transition-metal spin sublattices to orient differently; this results in a total of 36 different spin configurations.

We work with two model systems, GdFeO$_3$ and GdCrO$_3$, which are particularly well-behaved as regards non-collinear magnetic simulations performed with our computational tools (see Methods). Let us stress that this choice does not imply any loss of generality: We do not restrict our study to the specific magnetic ground state of the simulated gadolinium compounds, but rather consider all the spin configurations that are potentially relevant to the whole orthoferrite and orthochromite families. Further, the electric polarization that we expect to observe will be driven by the symmetry breaking imposed by the spin structure, possibly in combination with structural distortions that are qualitatively identical in all orthoferrites and orthochromites. Hence, consideration of GdFeO$_3$ and GdCrO$_3$ allows us to investigate all the qualitative effects that the *Pbnm* compounds can potentially display, and to draw general rules and conclusions. Needless to say, quantifying in detail the behaviour of particular compounds would require a specific investigation.

For each of the 36 considered spin arrangements, we proceed in the following way: we start the first-principles calculation with the atoms in the ideal (high-symmetry) positions of the *Pbnm* phase and the Gd and Fe/Cr magnetic moments initialized according to the specific spin arrangement. Then, we run a self-consistent field density functional theory (DFT) calculation (allowing for non-collinear magnetism and including spin–orbit interactions; see Methods) in which the spins are free to optimize the energy. The self-consistent field calculations are repeated along a full structural relaxation that stops when the forces on the atoms and stresses on the cell fall below a small threshold (see Methods). Finally, the resulting configuration is analysed, focusing on the optimized spin structure and the possible occurrence of an electric polarization (see Methods for details on the calculation of localized magnetic moments and polarization).

Table 1 summarizes the magnetic orders and polar distortions observed in the 36 cases considered, as obtained from our numerical simulations and verified by a symmetry analysis. Interestingly, only 13 cases, out of the 36 ones considered here, were numerically found to result in a polar solution. Table 2 details the spin arrangement obtained for two such polar cases.

The results for GdCrO$_3$ regarding the existence or absence of an electrical polarization are qualitatively identical to those

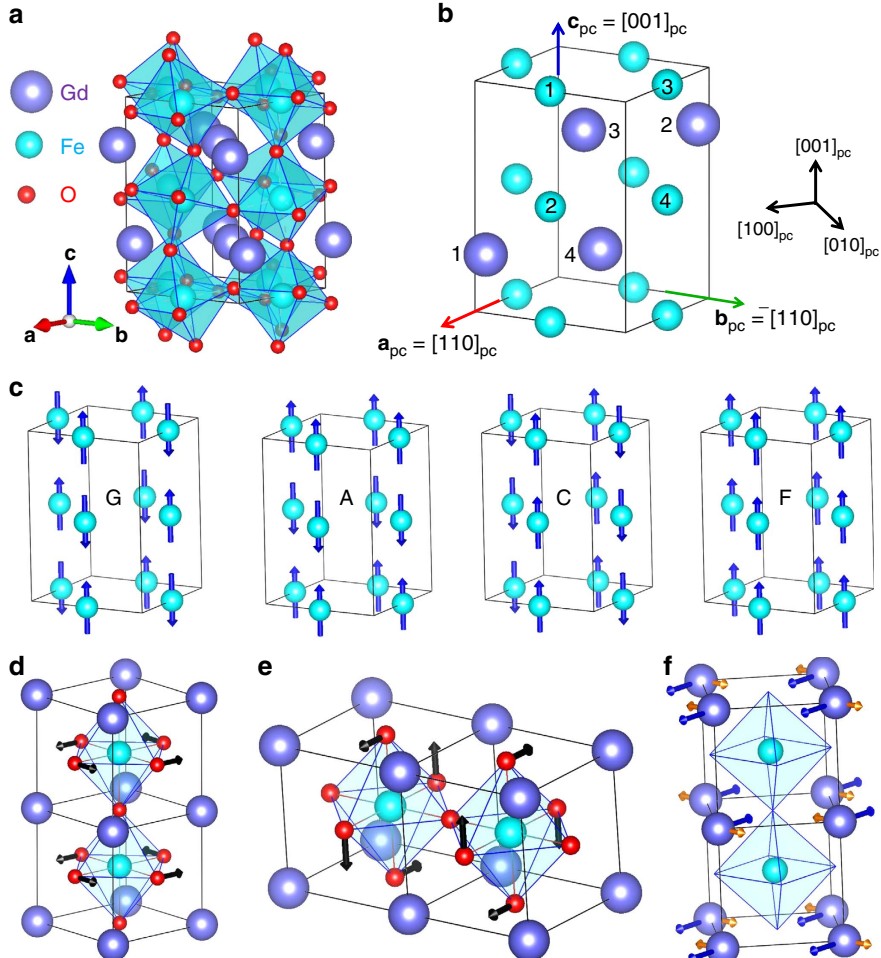

**Figure 1 | Most relevant crystal distortions and spin configurations in orthoferrites and orthochromites.** (**a**) Sketch of the perovskite structure in the 20-atom cell used in our calculations of GdFeO$_3$ and GdCrO$_3$. (**b**) Definition of the orthorhombic (*a*, *b* and *c*) and pseudo-cubic (*a*$_{pc}$, *b*$_{pc}$ and *c*$_{pc}$) axes. The rare-earth and transition-metal atoms are numbered. (**c**) Basic spin arrangements considered in this study (note that 'F' stands for ferromagnetic order). We indicate them for the Fe/Cr sublattice, those of the rare-earth sublattice being analogous. Sketches of atomic distortions discussed in the text: (**d**) in-phase O$_6$ rotations, (**e**) antiphase O$_6$ rotations and (**f**) anti-polar motions. In **f**, the larger arrows correspond to an *X*-point anti-polar modulation vector, while the smaller arrows indicate an *R*-point modulation (see text).

obtained for GdFeO$_3$, and are summarized in Table 3. This overall coincidence between the GdFeO$_3$ and GdCrO$_3$ results strongly supports our hypothesis that discussing these two specific compounds is relevant to understand the origin of electric polarization in all orthoferrites and orthochromites. For the sake of simplicity, in the following we will focus our discussion on the results for GdFeO$_3$.

Let us note that we also ran simulations in which the Gd$^{3+}$ cations were treated as non-magnetic (with their 4*f* electrons frozen at the ionic core), and explicitly checked that no polar distortion is obtained in absence of Gd-magnetism. This further demonstrates that the existence of a magnetic ordering at both the rare-earth and Fe/Cr sublattices is necessary to obtain an electric polarization, and is in agreement with the symmetry analysis made by various authors[18–20].

Interestingly, the obtained polar solutions can be readily classified in two categories. First, we have those in which the rare-earth spins have a dominant G-AFM order parallel to the G-AFM vector of the Fe sublattice. These three cases display a relatively large polarization. In addition, we have polar solutions in which the *R* spins present dominant G-AFM (four cases) or A-AFM (six cases) orders, with the corresponding AFM vectors being orthogonal to the G-AFM vector for the Fe sublattice. As

indicated in Table 1, in these situations we obtain very small polarization values (about 50–100 times smaller than those computed for the first category of polar solutions) that fall below the accuracy of our calculations. Let us first analyse in some detail the former category, which will be the focus of this paper; we will conclude this section with a brief discussion of the latter.

We get the largest polarization values for the cases sketched in Fig. 2. In all three cases the Gd and Fe spins exhibit a G-AFM order and the corresponding AFM vectors are parallel to each other; more precisely, they both lie along the *a* direction in the case depicted in Fig. 2a, along b in the case of Fig. 2b, and along c in Fig. 2c. In all three cases the atomic symmetry is orthorhombic *Pna2$_1$*, with the polar axis being along the *c* direction. Note that this common symmetry only pertains to the atomic positions; obviously, the three different cases correspond to three different magnetic space groups, as detailed in Table 1. As also shown in the table, in all the cases the polarization is predicted to be around 0.4 μC cm$^{-2}$; this is of the same order of magnitude as the experimental value of 0.12 μC cm$^{-2}$ reported in ref. 15 for GdFeO$_3$ at 2 K, and therefore about two orders of magnitude smaller than those of classic ferroelectrics such as BaTiO$_3$ or PbTiO$_3$. (We explicitly checked that the computed polarization value depends weakly on the Hubbard-*U* correction used in the

**Table 1 | Spin arrangements investigated for GdFeO₃**

| Gd spins | Fe spins | | |
|---|---|---|---|
| | $G_a$ | $G_b$ | $G_c$ |
| $G_a$ | $Pn'a'2_1$<br>Fe ($\mathbf{G}_a$, $A_b$, $F_c$)<br>Gd ($\mathbf{G}_a$, $A_b$, $F_c$)<br>$P_c = -0.42$ | $P2_12_12_1$<br>Fe ($A_a$, $\mathbf{G}_b$, $C_c$)<br>Gd ($\mathbf{G}_a$, $A_b$, $C_c$)<br>$P = 0$ | $P2_1n'm'$<br>Fe ($F_a$, $C_b$, $\mathbf{G}_c$)<br>Gd ($\mathbf{G}_a$, $C_b$, 0)<br>$P_a < 0.01$ |
| $G_b$ | $P2_1'2_12_1$<br>Fe ($\mathbf{G}_a$, $A_b$, $F_c$)<br>Gd ($A_a$, $\mathbf{G}_b$, $F_c$)<br>$P = 0$ | $Pna2_1$<br>Fe ($A_a$, $\mathbf{G}_b$, $C_c$)<br>Gd ($A_a$, $\mathbf{G}_b$, $C_c$)<br>$P_c = -0.41$ | $Pb2_1m'$<br>Fe ($F_a$, $C_b$, $\mathbf{G}_c$)<br>Gd ($F_a$, $\mathbf{G}_b$, 0)<br>$P_b < 0.01$ |
| $G_c$ | $P2_1'n'm$<br>Fe ($\mathbf{G}_a$, $A_b$, $F_c$)<br>Gd (0, 0, $\mathbf{G}_c$)<br>$P_a < 0.01$ | $Pb2_1m$<br>Fe ($A_a$, $\mathbf{G}_b$, $C_c$)<br>Gd (0, 0, $\mathbf{G}_c$)<br>$P_b < 0.01$ | $Pn'a2_1'$<br>Fe ($F_a$, $C_b$, $\mathbf{G}_c$)<br>Gd ($F_a$, $C_b$, $\mathbf{G}_c$)<br>$P_c = -0.41$ |
| $A_a$ | $P2_1'2_12_1$<br>Fe ($\mathbf{G}_a$, $A_b$, $F_c$)<br>Gd ($\mathbf{A}_a$, $G_b$, $F_c$)<br>$P = 0$ | $Pna2_1$<br>Fe ($A_a$, $\mathbf{G}_b$, $C_c$)<br>Gd ($\mathbf{A}_a$, $G_b$, $C_c$)<br>$P_c < 0.01$ | $Pb2_1m'$<br>Fe ($F_a$, $C_b$, $\mathbf{G}_c$)<br>Gd ($\mathbf{A}_a$, $C_b$, 0)<br>$P_b < 0.01$ |
| $A_b$ | $Pn'a'2_1$<br>Fe ($\mathbf{G}_a$, $A_b$, $F_c$)<br>Gd ($G_a$, $\mathbf{A}_b$, $F_c$)<br>$P_c < 0.01$ | $P2_12_12_1$<br>Fe ($A_a$, $\mathbf{G}_b$, $C_c$)<br>Gd ($G_a$, $\mathbf{A}_b$, $C_c$)<br>$P = 0$ | $P2_1n'm'$<br>Fe ($F_a$, $C_b$, $\mathbf{G}_c$)<br>Gd ($F_a$, $\mathbf{A}_b$, 0)<br>$P_a < 0.01$ |
| $A_c$ | $Pb'2_1'm$<br>Fe ($\mathbf{G}_a$, $A_b$, $F_c$)<br>Gd (0, 0, $\mathbf{A}_c$)<br>$P_b \sim 0.01$ | $P2_1nm$<br>Fe ($A_a$, $\mathbf{G}_b$, $C_c$)<br>Gd (0, 0, $\mathbf{A}_c$)<br>$P_a < 0.01$ | $P2_12_1'2_1'$<br>Fe ($F_a$, $C_b$, $\mathbf{G}_c$)<br>Gd ($F_a$, $C_b$, $\mathbf{A}_c$)<br>$P = 0$ |
| $C_a$ | $P2_1'/c'$ | $P2_1/c$ | $P2_1'/m'$ |
| $C_b$ | $P2_1'/c'$ | $P2_1/c$ | $Pbn'm'$ |
| $C_c$ | $P2_1/m$ | $Pbnm$ | $P2_1/c$ |
| $F_a$ | $P2_1'/c'$ | $P2_1/c$ | $Pbn'm'$ |
| $F_b$ | $P2_1'/c'$ | $P2_1/c$ | $P2_1'/m'$ |
| $F_c$ | $Pb'n'm$ | $P2_1/m$ | $P2_1'/c'$ |

The G, A, C and F letters indicate the presence of the spin order thus denoted (see text). The a, b and c subindexes indicate the direction of the corresponding spin order in the orthorhombic setting (see sketch in Fig. 1b). The spin arrangement used to initialize the simulation is indicated in bold font; the other (canted) orders appear in our calculations as a result of the symmetry breaking. (We only report magnetic-order components for which we obtain a sizeable result from our numerical calculations; other symmetry-allowed orders may in principle exist in some of the cases considered.) Magnetic space groups are given. In the cases in which the symmetry is polar, the polarization (computed using the Berry-phase formalism; see Methods) is indicated in μC cm⁻². Note that in some instances the polarization is very small and falls below the accuracy of our numerical method (see text).

**Table 2 | Detail of relevant spin configurations.**

| Atom | $G_a/G_a$ config. | | | $G_b/G_b$ config. | | |
|---|---|---|---|---|---|---|
| | $m_a$ | $m_b$ | $m_c$ | $m_a$ | $m_b$ | $m_c$ |
| Gd(1) | 6.948 | 0.003 | 0.011 | 0.002 | 6.948 | − 0.007 |
| Gd(2) | − 6.948 | − 0.002 | 0.011 | − 0.001 | − 6.948 | − 0.007 |
| Gd(3) | 6.948 | − 0.002 | 0.011 | − 0.002 | 6.948 | 0.007 |
| Gd(4) | − 6.948 | 0.002 | 0.011 | 0.002 | − 6.948 | 0.007 |
| Fe(1) | 4.053 | 0.021 | 0.023 | − 0.019 | 4.054 | − 0.015 |
| Fe(2) | − 4.053 | − 0.021 | 0.022 | 0.019 | − 4.054 | − 0.015 |
| Fe(3) | − 4.053 | 0.021 | 0.022 | − 0.019 | − 4.054 | 0.015 |
| Fe(4) | 4.053 | − 0.021 | 0.023 | 0.019 | 4.054 | 0.015 |
| Total | 0.000 | 0.000 | 0.141 | 0.000 | 0.000 | 0.000 |

These correspond to having dominant $G_a/G_a$ and $G_b/G_b$ orders for the Gd/Fe spin sublattices. Magnetic moments given in units of Bohr magneton ($\mu_B$) and estimated from our calculations as indicated in Methods. Gd and Fe atoms are numbered as in Fig. 1b. Note that the total magnetic moments include contributions from the oxygen atoms.

**Table 3 | Largest polarizations obtained.**

| Spin config. | $G_a/G_a$ | $G_b/G_b$ | $G_c/G_c$ |
|---|---|---|---|
| Polarization (μC cm⁻²) | − 0.14 | − 0.15 | − 0.14 |

Computed polarization $P_c$ of GdCrO₃ in the collinear Gd/Fe spin arrangements (see text).

simulations; more especifically, we obtain 0.37 μC cm⁻² for $U_{Gd} = 5$ eV and $U_{Fe} = 4$ eV and 0.46 μC cm⁻² when we use the smaller corrections $U_{Gd} = 3$ eV and $U_{Fe} = 2$ eV).

In the case of GdCrO₃, the predicted polarization is of the order of tenths of μC cm⁻², as well (Table 3), which is consistent with the experimental result of 0.6 μC cm⁻² reported in ref. 13. We should be cautious about this interpretation of the experimental data for GdCrO₃, though, as it is not clear whether the polarization measured in ref. 13 at relatively high-temperatures is a result of the simultaneous order of both spin sublattices or some other effect.

Table 1 further shows that when the G-AFM-ordered spins of both Gd and Fe sublattices lie parallel to the a axis, we also obtain a weak magnetization along c, which is consistent with the experimental observation of ref. 15 for GdFeO₃. In contrast, when

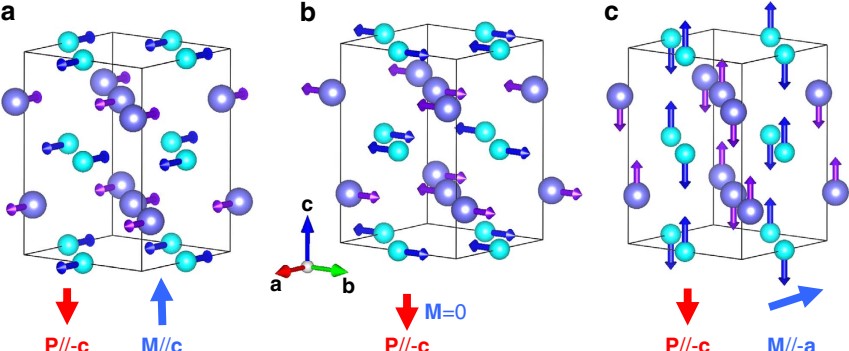

**Figure 2 | Spin configurations yielding the strongest ferroelectric polarizations.** In *a–c* the dominant G-AFM vectors are along *a*, *b* and *c* directions of *Pbnm* phases, respectively.

the Gd and Fe spins lie parallel to the *b* axis, no weak magnetization is observed. Finally, when the spins lie along *c*, the weak magnetization appears again, but is now antiparallel to the *a* axis. Hence, we find that the presence of a polar distortion (always along *c*) is independent from the existence or absence of a net magnetization, which is at odds with the picture proposed in refs 11,13. Further, in the cases in which a weak magnetization exists, the spins may cant parallel or perpendicular to the polarization axis. Note that a discussion of the canted spin orders in orthorhombic perovskites is given in refs 16,21.

It is thus clear that a sizable electric polarization appears in orthoferrites and orthochromites when both the *R* and Fe/Cr spin sublattices adopt a dominant G-AFM arrangment, with the AFM vectors being parallel to each other. The presence or absence of a weak magnetization is irrelevant as it regards the occurrence of the electric polarization. Further, the resulting polarization is largely independent from the direction along which the spins lie, which strongly suggests that it will occur even if we run our structural relaxations assuming that magnetization is described by a scalar field (that is, as in the standard non-relativistic calculations based on the local spin-density approximation or the so-called σ-generalized gradient approximation). Indeed, we explicitly verified this conjecture, obtaining *c*-oriented polarizations of about 0.40 and 0.16 μC cm$^{-2}$ for GdFeO$_3$ and GdCrO$_3$, respectively, which are essentially identical to those of Tables 1 and 3.

Let us now comment on the second category of polar solutions for which we found negligibly small polarization values. These occur when we have a dominant G-AFM or A-AFM order of the Gd spins, the corresponding AFM vector being orthogonal to that characterizing the G-AFM arrangement of the Fe sublattice. Hence, in this case the magnetically driven ferroelectricity fundamentally relies on having a non-collinear spin arrangement. Indeed, we explicitly checked that, for the case of Gd spins with A-AFM order, the effect disappears if we treat magnetism at the scalar level. (For Gd spins with G-AFM order, the scalar-magnetism calculation reduces to the one discussed above and yields a relatively large polarization). The reliance on a non-collinear spin structure, and the likely importance of spin–orbit interactions in many of these cases, probably explains the very small magnitude of the effect.

Hence, this initial exercise has shown that there are two possible routes to obtain a magnetically driven polar distortion in orthoferrites and orthochromites. The first one relies on a mechanism that is active at the scalar-magnetism level and in the absence of relativistic interactions; this mechanism leads to relatively large polarizations. The second one fundamentally relies on having a non-collinear magnetic configuration, and leads to

much smaller effects. In the following we focus our discussion on the first mechanism, which is most likely to explain the experimental data on GdFeO$_3$ and GdCrO$_3$ and, because it is stronger, can be expected to offer more interesting possibilities as regards the design or optimization of magnetoelectric multiferroics.

**Structural underpinnings.** To identify the origin of ferroelectricity in these materials, let us first indicate that the induced polar distortions mainly involve the rare-earth cations, which spontaneously move out of their positions in the centro-symmetric *Pbnm* state as the Gd and Fe/Cr spins order in collinear G-AFM patterns. Further, it can be checked numerically that the spin order by itself is not enough to explain the onset of ferroelectricity. Indeed, when we run a simulation with G-AFM-ordered Gd and Fe/Cr spins, while maintaining the atoms at the positions corresponding to the ideal *cubic* perovskite structure, we obtain no indication that a ferroelectric distortion will appear. (One such indication would be, for example, forces of the appropriate (polar) symmetry acting on the atoms. This kind of signature is further discussed and used below). It is thus clear that the spin arrangement somehow cooperates with the structural distortions characterizing the orthorhombic structure to produce the symmetry breaking that leads to the electric polarization. Further, the strength of the corresponding magnetostructural couplings will determine the magnitude of the induced ferroelectric distortion. Note that such couplings will be of the so-called exchange-strictive type, as our simulations indicate that they occur even in absence of relativistic spin–orbit interaction.

To identify which structural distortions mediate the occurrence of a magnetically driven polarization in orthoferrites and orthochromites, we run the following computational experiments. In what follows the Gd and Fe spins are always G-AFM-ordered and we treat magnetism at the scalar level. Let us focus on the calculations done for GdFeO$_3$, the results for GdCrO$_3$ being qualitatively identical. First, we consider GdFeO$_3$ in the ideal cubic perovskite structure, assuming a cubic lattice constant obtained by averaging the experimental values for the orthorhombic phase. Because of the cubic symmetry, the forces on the atoms are strictly null in this configuration. Then, we distort this reference state following each of the symmetry-inequivalent atomic distortions that characterize the *Pbnm* ground state. We compute the forces induced on the Gd and Fe atoms as we condense the different distortion modes, and then calculate the average force on the Gd and Fe sublattices; note that these polar forces act as (that is, have the symmetry of) an effective electric field and drive the onset of the improper ferroelectric

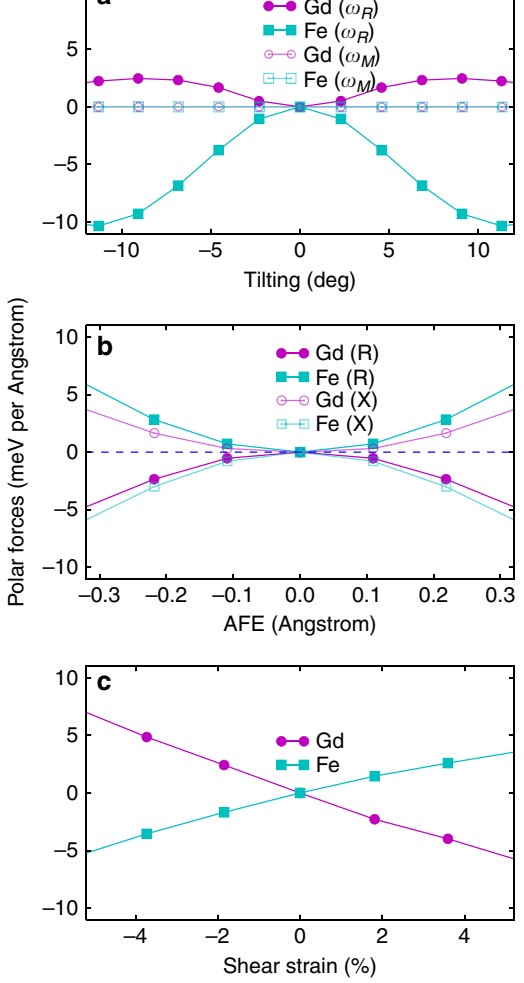

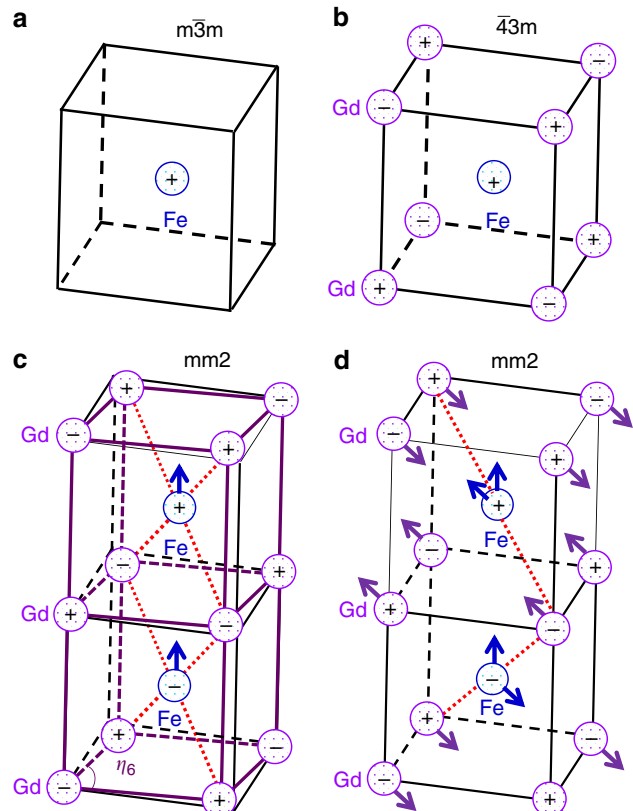

**Figure 4 | Representative cases of symmetry breaking and onset of improper polarization.** Sketches illustrating the symmetry-breaking caused by having G-AFM spin configurations in both the rare-earth (Gd) and transition-metal (Fe) sublattices. In **a** only magnetic order within the Fe sublattice is assumed, while in the other panels both sublattices are ordered. **c** illustrates the symmetry breaking associated to the $\eta_6$ shear strain, and in **d** we sketch $X$-point modulated anti-polar motions of the Gd cations. To better visualize the symmetry breaking, in **c,d** we use dashed lines connecting the Fe–Gd pairs that get closer because of the distortions, and arrows on the Fe cations indicate the induced forces. In addition, note that in **c,d** we show two elemental perovskite cells, so that the spatial modulation of the induced forces can be easily appreciated.

**Figure 3 | Polar forces leading to the improper ferroelectric polarization.** Average polar forces acting on the Gd and Fe atom sublattices, calculated as a function of various distortions of a reference (cubic) perovskite structure, in the scalar-magnetism approximation and assuming that both spin sublattices are G-AFM ordered (see text for more details). Results as a function of $O_6$ rotations (**a**), anti-polar distortion patterns (**b**) and shear strain (**c**). Circles and squares correspond to the results for Gd and Fe, respectively. In **a**, solid and open symbols correspond to antiphase and in-phase rotations, respectively. In **b**, solid and open symbols correspond to $R$-point and $X$-point modulated anti-polar patterns, respectively. For reference, note that our relaxed (the experimental) orthorhombic GdFeO$_3$ structure is characterized by antiphase and in-phase tilting angles of 11.99° (11.38°) and 11.91° (11.95°), respectively; anti-polar distortions of 0.092 Å (0.083 Å) and 0.369 Å (0.353 Å), respectively, for the $R_5^+$ and $X_5^+$ modes; and a shear strain of about 5.1% (4.8%).

polarization. Our result for the evolution of the Gd and Fe polar forces as a function of the various distortion modes is shown in Fig. 3.

Figure 3a shows the results for the dominant distortions characterizing the orthoferrite and orthochromite compounds, that is, the antiphase (about the *a* axis) and in-phase (about *c*) $O_6$-octahedra rotations that determine the symmetry breaking from cubic to orthorhombic and constitute the primary order parameters of the structural transition (Fig. 1d,e). Such a $O_6$-rotational pattern is termed $a^-a^-c^+$ in the notation introduced by Glazer[22]. Our results show that the antiphase rotations create a considerable polar force on the Gd atoms, while

the in-phase rotations do not induce any force at all. Interestingly, the dependence of the polar force on the antiphase rotation amplitude is quadratic, with a quartic behaviour appearing for large distortions. For reference, the amplitude of the various distortions in the equilibrium orthorhombic phase of GdFeO$_3$ is indicated in the caption of Fig. 3.

Then, we have large and very important secondary distortions involving anti-polar displacements of the rare-earth cations in the *ab* plane (Fig. 1f). There are two such anti-polar modes, corresponding respectively to the $X$ ($2\pi/a_{pc}(0, 0, 1/2)$) and $R$ ($2\pi/a_{pc}(1/2, 1/2, 1/2)$) $q$-points of the first Brillouin zone of the ideal cubic perovskite phase, where $a_{pc}$ is the pseudo-cubic lattice parameter. The polar forces associated to these modes are shown in Fig. 3b. The dependence of the force on the distortion amplitude is quadratic for both anti-polar patterns.

Finally, the cell strains in the cubic to orthorhombic transformation, the most important effect being associated to the $\eta_6$ shear controlling the angle formed by the $\mathbf{a}_{pc}$ and $\mathbf{b}_{pc}$ pseudo-cubic lattice vectors (defined in Fig. 1b). Figure 3c shows the corresponding polar forces, which have a linear dependence with the shear strain amplitude.

We thus find a remarkable result, namely, that several (namely, four) distortions of the *Pbnm* structure contribute to the onset of the magnetically driven electric polarization in these orthoferrites and orthochromites.

**Key magnetostructural couplings**. To gain further insight into the atomistic underpinnings of these couplings, it is most convenient to start with the last one of the cases mentioned above, that is, the linear dependence of the polar force (and, hence, of the induced polarization) with a shear deformation of the cell. This is a striking result with very broad implications: It tells us that any cubic $ABO_3$ perovskite with magnetic $A$ and $B$ sublattices will behave as a piezoelectric (that is, a material in which elastic strain induces an electric polarization) if the $A$ and $B$ spins acquire a (collinear) G-type AFM order.

Figure 4 sketches the symmetry breaking that drives such an interesting effect. Indeed, it is immediate to realize that, when magnetically disordered, the cubic perovskite phase is centro-symmetric and, in particular, the Gd and Fe/Cr cations are sitting at inversion centres of the lattice. Further, if only the Fe/Cr spins order (Fig. 4a), the structure is still centro-symmetric (point group $m\bar{3}m$). More precisely, assuming that spatial inversion does not change the sign of the atomic magnetic moments, from Fig. 4a we can see that the Fe/Cr atoms continue to occupy inversion centres, while at Gd sites the inversion symmetry is recovered when combined with a time reversal transformation. (The time reversal operation changes the sign of all the spins, but has no effect on the atomic structure). In contrast, when both the Gd and Fe/Cr spins are G-AFM-ordered (Fig. 4b), all the inversion-symmetry points are lost, and cannot be recovered even if we combine the spatial-inversion operation with time reversal. Hence, in this case the cubic phase is non-centro-symmetric (point group $\bar{4}3m$) and will present a piezoelectric response, even if it is non-polar. Indeed, as shown in Fig. 4c, when we apply a shear strain in the *ab* plane of such a structure, the environment of the Fe cations becomes asymmetric along the *c* direction, which guarantees that *c*-oriented forces acting on the Fe atoms will appear. This is a typical exchange-strictive effect: the spin-up Fe atom in Fig. 4 will surely have a preference between bonding to the two spin-up Gd atoms that approach it from above, or with the two spin-down Gd's that come closer from below; thus the polar force appears. As sketched in the figure, it is easy to demonstrate that this symmetry breaking occurs in exactly the same way everywhere in the lattice, thus resulting in an homogeneous force field that gives rise to an homogeneous effective electric field. This is thus the origin of the strain $\eta_6$-driven polar force of Fig. 3c.

Mathematically, we can model this effect by deriving the invariant coupling that appears in a Landau potential describing the energetics of the material around the reference cubic phase as a function of the relevant macroscopic order parameters. Such order parameters are the two G-AFM orders in the system as quantified by scalars $G_{Gd}$ and $G_{Fe}$, the polarization $P_\alpha$, and the shear strain $\eta_{\alpha\beta}$; here, $\alpha$ and $\beta$ label the Cartesian directions that we assume coincide with the pseudo-cubic lattice vectors of the perovskite structure. Then, the magnetostructural (ms) coupling leading to the effect in Fig. 3c is given by

$$E_{shear}^{ms} = K_{shear}^{ms} G_{Gd} G_{Fe} \sum_{\alpha\beta\gamma} |\epsilon_{\alpha\beta\gamma}| \eta_{\alpha\beta} P_\gamma, \qquad (1)$$

where $\epsilon_{\alpha\beta\gamma}$ is the Levi-Civita symbol, which equals $+1$ when the ordered triad $\alpha\beta\gamma$ forms a right-handed system, $-1$ when left-handed, and 0 when there are repeated indexes. In addition, $\eta_{\alpha\beta}$ (for $\alpha \neq \beta$) denotes the symmetric part of a shear strain; in

particular we have $2\eta_{xy} = \eta_6$ in Voigt notation. Finally, note that we can replace the product $G_{Gd} G_{Fe}$ by the dot product $\mathbf{G}_{Gd} \cdot \mathbf{G}_{Fe}$ if we want to recover a three-dimensional representation of the magnetism.

Let us note a subtlety concerning the invariance of such a coupling term with respect to the symmetry operations of the cubic reference phase. If we think of the action of space inversion on such a term, for the structural order parameters we trivially have $P_\alpha \to -P_\alpha$ and $\eta_{\alpha\beta} \to \eta_{\alpha\beta}$, without any ambiguity. In contrast, the transformation of the magnetic order parameters depends on the precise location of the inversion centre that we consider. Thus, for example, if we consider a Fe-centred inversion point, we have $G_{Gd} \to -G_{Gd}$ and $G_{Fe} \to G_{Fe}$; in contrast, if we consider inversion points at the Gd or oxygen sites, we get $G_{Gd} \to G_{Gd}$ and $G_{Fe} \to -G_{Fe}$. Nevertheless, in all cases it is true that the product $G_{Gd} G_{Fe}$ changes sign upon spatial inversion, as needed for the $E_{shear}^{ms}$ term to be invariant. Note that this dependence on the location of the inversion point is not related to the magnetic character of the $G_{Fe}$ and $G_{Gd}$ order parameters, but to the fact that they are associated to a *q*-point different from $\Gamma$. In fact, similar considerations apply to the $O_6$-rotational order parameters discussed below.

Note that the $E_{shear}^{ms}$ coupling just described is linear in the polarization, and will thus lead to an effective electric field of the form:

$$\mathcal{E}_{shear,\alpha}^{ms} \sim -K_{shear}^{ms} G_{Gd} G_{Fe} \sum_{\beta\gamma} |\epsilon_{\alpha\beta\gamma}| \eta_{\beta\gamma}. \qquad (2)$$

In particular, this implies that the $\eta_6 = 2\eta_{xy}$ shear will induce an effective field, and thus an improper polarization, along *c*, in agreement with our DFT results. (Note that such an effective field will induce proportional polar forces acting on the various atomic sublattices). In addition, in agreement with our results for the polar forces of Fig. 3c, both the effective field and the induced polarization will be linear in $\eta_6$.

Similarly, it is possible to identify the symmetry-invariant couplings responsible for the other effects shown in Fig. 3a,b. More precisely, we have

$$E_{rot-R}^{ms} = K_{rot-R}^{ms} G_{Gd} G_{Fe} \sum_{\alpha\beta\gamma} |\epsilon_{\alpha\beta\gamma}| \omega_{R\alpha} \omega_{R\beta} P_\gamma, \qquad (3)$$

$$E_{ap-R}^{ms} = K_{ap-R}^{ms} G_{Gd} G_{Fe} \sum_{\alpha\beta\gamma} |\epsilon_{\alpha\beta\gamma}| A_{R\alpha} A_{R\beta} P_\gamma, \qquad (4)$$

and

$$E_{ap-X}^{ms} = K_{ap-X}^{ms} G_{Gd} G_{Fe} \sum_{\alpha\beta\gamma} |\epsilon_{\alpha\beta\gamma}| A_{X_\gamma\alpha} A_{X_\gamma\beta} P_\gamma, \qquad (5)$$

where $\omega_{R\alpha}$ denotes the amplitude of antiphase $O_6$ rotations about the pseudo-cubic axis $\alpha$; $A_{R\alpha}$ and $A_{X_\gamma\alpha}$ denote anti-polar distortions along $\alpha$ and modulated according to $R$ and $X$-type *q*-points of the Brillouin zone, respectively; in the case of $A_{X_\gamma\alpha}$, the $X$-type point is $2\pi/a_{pc}(1/2, 0, 0)$ for $\gamma = x$, $2\pi/a_{pc}(0, 1/2, 0)$ for $\gamma = y$ and $2\pi/a_{pc}(0, 0, 1/2)$ for $\gamma = z$. All these terms yield effective electric fields that are quadratic in the amplitude of the structural distortions, in agreement with our numerical results for the polar forces of Fig. 3a,b. As an illustrative example, Fig. 4d illustrates the occurrence of homogeneous polar forces caused by the interactions captured by equation (5). Note also that couplings with a quartic dependence of these structural distortions can be similarly deduced.

It is also possible to demonstrate that, in contrast, symmetry precludes the existence of couplings that would lead to a linear dependence of the polar forces on the antiphase rotational or anti-polar order parameters. For example, a coupling of the form $\sim G_{Gd} G_{Fe} P_\alpha \omega_{R\beta}$ is incompatible with translational symmetry

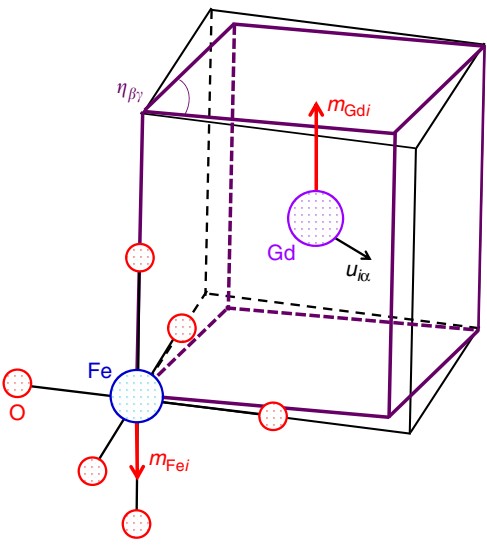

**Figure 5 | Relevant atomistic variables.** Sketch of the atomistic variables involved in the magnetostructural coupling described by equation (6).

(among others) and cannot exist. Again, this is compatible with our numerical findings. Similarly, symmetry precludes the onset of polar forces as in-phase rotations condense on their own, as consistent with the null result shown in Fig. 3a.

To finish this part, let us note that the above energy couplings, which involve macroscopic order parameters, can ultimately be deduced from atomistic interactions, that is, interactions involving atomic spins, off-centring displacements of specific Gd and Fe cations, rotations of individual $O_6$ octahedra, and so on. There are recent examples of such an atomistic approach to investigate these and similar perovskite materials, for example, in what regards the interplay between rotational and anti-polar distortions[23] or the coupled magnetic orders and origin of weak magnetization in orthoferrites and orthochromites[16,21]. Working with atomistic couplings is also a necessary step towards the development of effective models—for example, of the so-called effective Hamiltonian type[24–26]—that can eventually permit large-scale statistical simulations of these multi-ferroic effects. Here, to illustrate such an approach, we describe the simplest atomistic coupling that leads to the $E_{shear}^{ms}$ energy discussed above.

As sketched in Fig. 5, let $u_{i\alpha}$ denote the off-centring displacement of the Gd cation at cell $i$ along Cartesian direction $\alpha$, and let $m_{Gd,i}$ be the (scalar) magnetic moment associated to the same Gd cation. Also, let $m_{Fe,ilmn}$ be the magnetic moment of the Fe cation located at the cell defined by $\mathbf{R}_i + l\mathbf{a}_{pc} + m\mathbf{b}_{pc} + n\mathbf{c}_{pc}$, where $\mathbf{R}_i$ is the lattice vector corresponding to cell $i$; $\mathbf{a}_{pc}$, $\mathbf{b}_{pc}$, and $\mathbf{c}_{pc}$ are the lattice vectors corresponding to the five-atom pseudo-cubic perovskite cell; and $l$, $m$ and $n$ are integers. Finally, let $\eta_{\alpha\beta}$, for $\alpha \neq \beta$, be the symmetric part of the homogeneous strain tensor, which we use to represent the strain state of individual cells, as shown in Fig. 5. Then, if we search for the simplest symmetry-allowed coupling involving the Gd and Fe spins, as well as the Gd off-centring distortion and cell shear strains, we obtain

$$E_{shear}^{ms,at} = K_{shear}^{ms,at} \sum_i \sum_{l,m,n=0,1} \sum_{\alpha\beta\gamma} \left|\epsilon_{\alpha\beta\gamma}\right| (-1)^{l+m+n}$$
$$\times u_{i\alpha}\eta_{\beta\gamma}m_{Gdi}m_{Feilmn}, \qquad (6)$$

where $K_{shear}^{ms,at}$ is a material-dependent-coupling constant. Now, let us consider the particular case in which both the Gd and Fe spin sublattices adopt a G-type AFM order, so that $m_{Gdi} = G_{Gd}(-1)^{l_i + m_i + n_i}$ and $m_{Fei} = G_{Fe}(-1)^{l_i + m_i + n_i}$, where $\mathbf{R}_i = l_i\mathbf{a}_{pc} + m_i\mathbf{b}_{pc} + n_i\mathbf{c}_{pc}$. Let us also imagine that the Gd

off-centrings are homogeneous throughout the lattice, so that $u_{i\alpha} = u_\alpha \quad \forall i$; such distortions generate a polarization $P_\alpha = \Omega^{-1} Z_{Gd}^* u_\alpha$, where $Z_{Gd}^*$ is the relevant effective charge of the Gd cation and $\Omega$ is the volume of the five-atom perovskite cell. It is thus trivial to verify that the atomistic coupling in equation (6) leads to the macroscopic interaction of equation (1), with $K_{shear}^{ms} = 8\,\Omega K_{shear}^{ms,at}/Z_{Gd}^*$.

## Discussion

Let us now discuss the implications of our results from two perspectives, a general one that concerns multiferroic perovskites at large, and one focused on previous and current work on multiferroic orthoferrites and orthochromites.

In a generic way, our results can be described as examples of improper ferroelectricity driven by spin ordering. The magnitude of the induced electric polarization depends on magnetostructural couplings, with the strongest (exchange-strictive) effects corresponding to collinear spin configurations and present in the absence of relativistic interactions. Our findings support the essential correctness of previous discussions of magnetically induced polarization in these materials[6,10], which were based on empirically deduced models or simplified first-principles studies. Note that, at variance with the present work, the previous studies did not consider all the existing magnetostructural couplings, nor did they explain the effects with a detailed theory. Interestingly, let us mention that the collinear A-AFM orders hypothesized for both $R$ and $M$ spin sublattices in these references actually occur, as a weak-canted orders, in many of the spin configurations considered in our work (Table 1). From the point of view of a polarization induced by symmetry breaking, such cases are thus essentially identical.

Our results and analysis go one step further, as they offer valuable insights into the atomistic underpinnings of the magnetically driven ferroelectricity, as well as hints at how to optimize these effects or obtain related ones in similar situations. For example, let us recall the mechanism described by equation (1), by which a shear–strain mediates the occurrence of the magnetically driven polarization. The presence of such an interaction implies that it is possible to enhance the electric polarization of orthoferrites and orthochromites by growing them as thin films on orthorhombic substrates presenting a large shear strain. One example of such substrates is the commonly used $DyScO_3$ system for which $\eta_6 \approx 5.0\%$ (ref. 27).

Similarly, the magnetostructural coupling given by equation (3) should yield an improper polarization for other $O_6$-rotational patterns; in other words, its action is by no means exclusive to the $a^- a^- c^+$ structure typical of $Pbnm$ phases. We explicitly checked this by simulating a $a^- a^- a^-$ polymorph of $GdFeO_3$ with our first-principles methods. Typically, this rotational pattern results in the non-polar $R\bar{3}c$ space group. However, because of the combination of the rotational distortion and G-AFM order for both Gd and Fe spin sublattices, the atomic symmetry of such a structure is polar with the rhombohedral $R3c$ space group. This is precisely the effect reflected in equation (3), and it results in a polarization of about $0.2\,\mu C\,cm^{-2}$ when we fully relax $GdFeO_3$ in this hypothetical rhombohedral phase. We explicitly checked that, if we suppress the Gd spins, by freezing them into the ionic core, we obtain the non-polar $R\bar{3}c$ symmetry and a null polarization. Hence, this polar distortion is genuinely driven by the simultaneous magnetic order of the Gd and Fe spin sublattices. Hence, knowledge of the atomistic couplings responsible for the polarization in orthoferrites and orthochromites allows us to identify alternative possibilities, corresponding to completely different crystallographic symmetries, in which the same sort of magnetically driven polarization can exist.

**Table 4 | Rules to derive symmetry invariants.**

| Order Param. | $C_2$ [100]$_{pc}$ | $C_2$ [010]$_{pc}$ | $C_3^+$ [111]$_{pc}$ | $C_2$ [110]$_{pc}$ | $\bar{1}$ | $1'$ |
|---|---|---|---|---|---|---|
| $P_x$ | $P_x$ | $-P_x$ | $P_y$ | $P_y$ | $-P_x$ | $P_x$ |
| $P_y$ | $-P_y$ | $P_y$ | $P_z$ | $P_x$ | $-P_y$ | $P_y$ |
| $P_z$ | $-P_z$ | $-P_z$ | $P_x$ | $-P_z$ | $-P_z$ | $P_z$ |
| $\eta_{xy}$ | $-\eta_{xy}$ | $-\eta_{xy}$ | $\eta_{yz}$ | $\eta_{xy}$ | $\eta_{xy}$ | $\eta_{xy}$ |
| $\eta_{yz}$ | $\eta_{yz}$ | $-\eta_{yz}$ | $\eta_{xz}$ | $-\eta_{xz}$ | $\eta_{yz}$ | $\eta_{yz}$ |
| $\eta_{xz}$ | $-\eta_{xz}$ | $\eta_{xz}$ | $\eta_{xy}$ | $-\eta_{yz}$ | $\eta_{xz}$ | $\eta_{xz}$ |
| $\omega_{Rx}$ | $\omega_{Rx}$ | $-\omega_{Rx}$ | $\omega_{Ry}$ | $-\omega_{Ry}$ | $-\omega_{Rx}$ | $\omega_{Rx}$ |
| $\omega_{Ry}$ | $-\omega_{Ry}$ | $\omega_{Ry}$ | $\omega_{Rz}$ | $-\omega_{Rx}$ | $-\omega_{Ry}$ | $\omega_{Ry}$ |
| $\omega_{Rz}$ | $-\omega_{Rz}$ | $-\omega_{Rz}$ | $\omega_{Rx}$ | $\omega_{Rz}$ | $-\omega_{Rz}$ | $\omega_{Rz}$ |
| $A_{Rx}$ | $A_{Rx}$ | $-A_{Rx}$ | $A_{Ry}$ | $A_{Ry}$ | $-A_{Rx}$ | $A_{Rx}$ |
| $A_{Ry}$ | $-A_{Ry}$ | $A_{Ry}$ | $A_{Rz}$ | $A_{Rx}$ | $-A_{Ry}$ | $A_{Ry}$ |
| $A_{Rz}$ | $-A_{Rz}$ | $-A_{Rz}$ | $A_{Rx}$ | $-A_{Rz}$ | $-A_{Rz}$ | $A_{Rz}$ |
| $A_{X_x x}$ | $A_{X_x x}$ | $-A_{X_x x}$ | $A_{X_y y}$ | $A_{X_y y}$ | $-A_{X_x x}$ | $A_{X_x x}$ |
| $A_{X_x y}$ | $-A_{X_x y}$ | $A_{X_x y}$ | $A_{X_y z}$ | $A_{X_y x}$ | $-A_{X_x y}$ | $A_{X_x y}$ |
| $A_{X_x z}$ | $-A_{X_x z}$ | $-A_{X_x z}$ | $A_{X_y x}$ | $-A_{X_y z}$ | $-A_{X_x z}$ | $A_{X_x z}$ |
| $A_{X_y x}$ | $A_{X_y x}$ | $-A_{X_y x}$ | $A_{X_z y}$ | $A_{X_x y}$ | $-A_{X_y x}$ | $A_{X_y x}$ |
| $A_{X_y y}$ | $-A_{X_y y}$ | $A_{X_y y}$ | $A_{X_z z}$ | $A_{X_x x}$ | $-A_{X_y y}$ | $A_{X_y y}$ |
| $A_{X_y z}$ | $-A_{X_y z}$ | $-A_{X_y z}$ | $A_{X_z x}$ | $-A_{X_x z}$ | $-A_{X_y z}$ | $A_{X_y z}$ |
| $A_{X_z x}$ | $A_{X_z x}$ | $-A_{X_z x}$ | $A_{X_x y}$ | $A_{X_z y}$ | $-A_{X_z x}$ | $A_{X_z x}$ |
| $A_{X_z y}$ | $-A_{X_z y}$ | $A_{X_z y}$ | $A_{X_x z}$ | $A_{X_z x}$ | $-A_{X_z y}$ | $A_{X_z y}$ |
| $A_{X_z z}$ | $-A_{X_z z}$ | $-A_{X_z z}$ | $A_{X_x x}$ | $-A_{X_z z}$ | $-A_{X_z z}$ | $A_{X_z z}$ |
| $G_{Gd}$ | $G_{Gd}$ | $G_{Gd}$ | $G_{Gd}$ | $G_{Gd}$ | $G_{Gd}$ | $-G_{Gd}$ |
| $G_{Fe}$ | $G_{Fe}$ | $G_{Fe}$ | $G_{Fe}$ | $-G_{Fe}$ | $-G_{Fe}$ | $-G_{Fe}$ |

Transformation of the various order parameters discussed in the text under the action of the generators of the $m\bar{3}m$ point group and time reversal symmetry ($1'$). We indicate the axis of the two-fold ($C_2$) and three-fold ($C_3^+$) rotations; $\bar{1}$ is the spatial inversion. To write these transformation rules, we assume that the symmetry operations are centred on an $A$ site of the perovskite lattice (see discussion in main text). All labels and indicated directions refer to the pseudo-cubic (pc) setting.

The last example emphasizes the generality of our results: Because the newly identified couplings are active in any perovskite structure with $A$ and $B$ magnetic sublattices, they apply to (and can be used to our advantage in) a large variety of compounds beyond the ferrites and chromites. Indeed, such couplings can potentially play an active role, and lead to multiferroic phases, in other orthorhombic perovskites, the nickelate ($R$NiO$_3$), manganite ($R$MnO$_3$) and titanate ($R$TiO$_3$) families being obvious candidates. Further, the couplings unveiled in this work should also be relevant to essentially all perovskite phases hosting two spin sublattices, as perovskites are all but guaranteed to present shear, O$_6$-rotational and/or anti-polar distortions of some form. Further, such structural distortions can often be induced or controlled by a variety of means, chemical substitution and strain engineering being the most usual ones. Hence, the possibilities to find or induce ferroelectric order in perovskites with two magnetic sublattices, thanks to the couplings here described, seem essentially endless.

Interestingly, the magnetically driven piezoelectric response, and ensuing improper polarization, revealed in this work can be viewed as an example of a broader family of effects. Thus, for instance, it has been recently shown that spin order may result in a contribution to the polarization of a ferroelectric via a combination of magnetoelastic and piezoelectric couplings[28]. Similarly, it is possible to obtain a linear magnetoelectric response in materials that are both piezomagnetic and piezoelectric[29]. Nevertheless, while they involve similar ingredients, there is a critical difference between such effects and the one that we have introduced in this work: in the present case the material is not piezoelectric in absence of magnetic order; instead, piezo-electricity is magnetically-driven. Hence, we could say that the mechanisms discussed in refs 28,29 are typical of the so-called type I multiferroics[17], where ferroelectricity is a proper order, while the mechanisms discussed here would generally correspond to type II materials with an improper (magnetically induced) polar order.

Our present results are somewhat reminiscent of recent observations on materials like LaMn$_3$Cr$_4$O$_{12}$ (refs 30–32), which presents a dual G-type collinear antiferrmagnetism that results in an electric polarization[30]. However, beyond obvious structural and stoichiometric differences, it should be noted that in LaMn$_3$Cr$_4$O$_{12}$ the magnetically driven polar distortion relies on the spin–orbit couplings, at variance with the main effects discussed here.

Let us conclude by discussing the implications of our results as regards previous works on $R$FeO$_3$ and $R$CrO$_3$. As already mentioned, the reasonable numerical agreement between our computed polarizations and those experimentally measured for some orthoferrites and orthochromites suggests that the polar order in those compounds emerges from having nearly-collinear G-AFM arrangements in both the rare-earth and transition-metal sublattices. Further, as already mentioned above, our results clearly show that the occurrence of a ferroelectric polarization is unrelated to the presence or absence of a net (weak) magnetization, at variance from the picture proposed in refs 11,13. More specifically, in refs 11,13 both the electric polarization and weak magnetization are observed to simultaneously disappear at a spin reorientation transition, which seems to imply a causal connection between them. Our results, though, indicate that, while there may be a common cause for these two effects, it is not the case that one is necessary for the other to occur.

Also, let us note that various works on ferrites (like, for example, SmFeO$_3$ (ref. 12)) and chromites (like, for example, ErCrO$_3$ (ref. 13)) have reported electric polarizations occurring at temperatures well above the ordering transition for the rare-earth spins. However, according to our present results and to the symmetry analysis made by others[18–20], a simple AFM order of the Fe/Cr spins cannot produce such polarizations in absence of rare-earth magnetism; instead, a cycloidal arrangement—as, for example, the one occurring in prototypic compound TbMnO$_3$ (ref. 33)—is needed for a polarization to appear.

In summary, we have conducted a systematic theoretical investigation on the sources of improper magnetically driven ferroelectricity in rare-earth orthoferrite and orthochromite perovskites. Our work has confirmed that both the rare-earth and transition-metal spin sublattices need to order for the multiferroic effects to occur. Further, our calculations reveal several possible scenarios leading to the improper polar order; in particular, we find that the largest polarizations are driven by exchange-strictive mechanisms and correspond to simple co-linear spin arrrangments. Our results are in reasonable agreement with (and explain) available experimental data.

We have identified in detail the magnetostructural couplings responsible for the largest polarizations found. Interestingly, four structural distortions of the ideal cubic perovskite contribute: one shear strain, one antiphase $O_6$-rotational mode, and two anti-polar modes. The dependence is linear for the shear strain (an effect that can be seen as a magnetically induced piezoelectricity of the cubic perovskite structure) and quadratic for the other distortions. We have modelled these couplings in the framework of a general Landau-like potential that applies to any $ABO_3$ perovskite with two magnetic sublattices, and confirmed explicitly that similar effects can be obtained (and enhanced) in compounds belonging to other families and presenting other symmetries. In particular, our DFT simulations showed that our discovered magnetostructural coupling involving antiphase $O_6$ rotations can drive improper ferroelectricity in rhombohedral ($a^- a^- a^-$) perovskites (the transition is $R\bar{3}c \rightarrow R3c$). Indeed, because the most relevant non-polar distortions of the perovskite structure are present in orthoferrites and orthochromites, our study has allowed us to map out essentially all the possibilities to obtain effects of this kind in these compounds, its conclusions being general and virtually exhaustive. Our work thus highlights perovskites with two magnetic sublattices as a rich playground for novel magnetoelectric effects, and provides the basic cartography of the field.

## Methods

**First-principles calculations.** For the simulations we use DFT as implemented in the Vienna *ab-initio* Simulation Package within the framework of the projected augmented wave method[34,35]. The generalized gradient approximation with the Perdew and Wang parameterization[36] was employed, using a 'Hubbard $U$' correction[37] for a better treatment of Fe and Cr's $3d$ electrons and Gd's $4f$ electrons. The $U$'s are chosen to be 4.0, 3.0 and 3.0 eV for Gd, Fe and Cr, respectively, in line with previous works[10]. We solved explicitly for the following electrons: $4f^7 5s^2 5p^6 5d^1 6s^2$ for Gd, $3d^7 4s^1$ for Fe, $3d^5 4s^1$ for Cr, and $2s^2 2p^4$ for O. Note that the Gd atoms are allowed to order magnetically, since their $4f$ electrons are explicitly treated in the simulations; alternatively, one can assume a $3+$ ionization state and consider the $4f$ electrons frozen in the ionic core, an approximation that we adopt to run some tests described in the text. The electronic wave functions were represented in a plane-wave basis cut off at 500 eV. The reciprocal space integrals were computed using a Monkhorst–Pack $k$-point mesh of $6 \times 6 \times 4$, corresponding to the 20-atom orthorhombic cell which can be viewed as a $\sqrt{2} \times \sqrt{2} \times 2$ multiple of the elemental five-atom perovskite unit. Note that this is the primitive cell for a $a^- a^- c^+$ pattern of oxygen octahedra rotations, as expressed in Glazer's notation[22]. Note also that the shear strain $\eta_6$ in the pseudo-cubic representation can be obtained from the lattice constants of the orthorhombic cell as $\eta_6 = 2(b-a)/(b+a)$.

Spin–orbit couplings and non-collinear magnetic states are both considered in the calculations, unless otherwise specified. The structural optimizations are carried out until the forces acting on the atoms go below 0.005 eV Å$^{-1}$; we checked this is sufficient by running relaxations down to 0.001 eV Å$^{-1}$ in a few representative cases.

**Symmetry analysis.** The space groups of various configurations are identified using the FINDSYM code, which is able to tackle magnetic symmetries[38] (http://iso.byu.edu/iso/findsym.php). The Berry phase method[39] is used to calculate electric polarizations. We typically run the Berry phase calculations at the scalar-magnetism level, but using the structures obtained from non-collinear calculations including spin–orbit couplings; the polarizations thus computed were checked to be correct in a few representative cases for which we ran the Berry phase calculation at the non-collinear level with spin–orbit. We use the VESTA code[40] to draw most of the sketches that appear in the present work, and the tools in the

Bilbao Crystallographic Server to conduct some structural and symmetry analyses (http://www.cryst.ehu.es).

Finally, for the benefit of the interested reader, we list in Table 4 the transformations of the order parameters appearing in our energy terms (equations (1–5)) under the generators of the full cubic point group ($m\bar{3}m$) and time reversal. With this information at hand, the symmetry-allowed invariants can be deduced.

**Data availability.** The original data of this study can be obtained from the corresponding author upon request.

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

## Acknowledgements

This work was supported by FNR Luxembourg Grants FNR/P12/4853155/Kreisel (H.J.Z. and J.Í.) and INTER/MOBILITY/15/9890527 GREENOX (L.B. and J.Í.). L.B. also thanks the support of the Office of Basic Energy Sciences, under contract ER-46612. X.M.C. also thanks support from the National Natural Science Foundation of China (grant number 51332006) and the National Basic Research Program of China (grant number 2016YFA0300101). Most of the calculations were run at the Arkansas High Performance Computing Center.

## Author contributions

All authors contributed to conceive the work, which was mainly carried out by H.J.Z. (calculations) and J.Í. (models). All authors contributed to prepare the manuscript.

## Additional information

**Competing financial interests:** The authors declare no competing financial interests.

**Publisher's note**: 

