## [Peer Review File · Nature Communications]

Reviewers' comments:

Reviewer #1 (Remarks to the Author):

The authors studied the origin of magnetically-driven ferroelectric order in ABO₃ perovskite oxides with magnetic A and B cations that order in simple spin structures. They propose a magnetically-driven piezoelectric effect to explain the improper ferroelectricity. In my opinion, this is a rather new concept. Therefore, I would suggest the publication of this work in Nature Communications after the authors address the following comments:

(1) In Fig. 4, I would suggest the authors to add the (magnetic) point group of the three cases so that the reader can understand better the idea of magnetically-driven piezoelectric effect.

(2) In Phys. Rev. B 91, 100405(R) (2015), it was proposed that there exists an additional contribution to the electric polarization induced by the spin order in multiferroics due to the combined effects of magnetoelastic coupling and piezoelectric effect. In some sense, the magnetically-driven piezoelectric effect is closely related to the magnetically-driven ferroelectricity due to the piezoelectric effect. Thus, I would suggest the authors cite that work and discuss the similarity and difference between these related but different effects.

(3) Typo: Line 314 "In order to indentify"

Reviewer #2 (Remarks to the Author):

In this work, Zhao et al. performed very detailed DFT calculations on multiferroic RFeO₃ and RCrO₃. The underlying physics of magnetism-induced-polarization has been clearly illustrated. Their conclusion is that the exchange striction dominantly contributes to the polarization once the magnetic moments of R and Fe/Cr are collinear. The roles of octahedral rotations, anti-phase motion of R, and shear distortion have been clarified. In addition, phenomenological Landau energy items have been formulated. This work is well-done and will be physical interesting to the multiferroic community. I have following comments:

1. The title is too broad. The systems understudy only involve RFeO₃ and RCrO₃. Although the mechanisms revealed in the present work can be extended to some similar systems, it remains too broad to cover all multiferroics with two magnetic sublattices. For example, the RMn₂O₅, CaMn₇O₁₂, and LaMn₃Cr₄O₁₂ systems also own two magnetic sublattices, which own different stories.

2. Fig 4c is helpful to understand the exchange striction between Fe and R. However, according to Fig. 3a and 3b, even without the shear strain, considerable polar forces still exist, in the same order of magnitude with that in 3c. Then what's the physical mechanism for Fig. 3a ad 3b?

3. I agree with the exchange striction. But I'm suspected on the detailed microscopic origin. In lines 310-313, the authors argued that "they involve regular (super)exchange mechanisms". Is there any solid evidence to support the superexchange between Fe/Cr and R? I'm quite suspected on this point. The authors need to check the energy band structure. If their bands are far from each other in energy (very possible), the superexchange should be negligible. A hint to support my guess is that the polarization disappears if the lattice is fixed. Previous DFT studies on many type-II multiferroics with exchange striction revealed significant contribution from pure electronic part, which is a hint of superexchange involved. Considering this fact and the very small polarization (comparing with other exchange-strictive multiferroics), I guess the interaction between Fe/Cr and R is mainly the magnetic dipole-dipole interaction, instead of the real superexchange.

4. I noticed that the dual G-type collinear antiferromagnetism in recently found LaMn₃Cr₄O₁₂ also

contributed to a polarization. However, the underlying physical mechanism, involving the spin-orbit coupling, is quite different from the current one. The authors need to clarify this point.

5. The discussion on SmFeO₃ seems to be needless to me. The experimental result on SmFeO₃ are quite suspicious and negatively commented by following works. The physics proposed here can not say anything on SmFeO₃. So I suggest the authors to discard the related discussion.

Reviewer #3 (Remarks to the Author):

In this manuscript the authors scrutinise from state of the art DFT calculations the microscopic origin of the electric polarisation observed in rare-earth orthoferrites and orthochromites. They first consider all the possible non-collinear magnetic configurations of both rare-earth and transition metal cations in GdFeO₃ and compute the resulting polarisation. They found that 13 cases over 36 are indeed polar. Over the 13 cases they distinguish that two cases can be separated: Those requiring spin-orbit coupling and those requiring only exchange-striction effect. They observe that the later has the largest induced polarisation and it is the keystone of the present paper. Then, they prove that this mechanism is present when both rare-earth and transition-metal atoms adopt parallel G-type AFM vector and is independent of the presence of a weak ferromagnetic component.

After having analysed the magnetic origin of the induced polarisation the authors explore what are the structural key parameters that cooperate with the magnetism to induce the polarisation. Here they originally proved that the presence of antiphase rotation of the octahedra and a shear strain are mandatory to observe the induced polarisation. They then show from a phenomenological-like model how this magnetostructural order can appear in these perovskite structures to couple with the polarisation.

The paper is well written and, beside a few comments I'll give below, the results are sufficiently striking, novel and important to be accepted for publication in Nature Communication. Additionally the paper will represent a reference and high impact article in view of the present increase of interest for rare-earth based perovskites.

1- From line 150 to 155, the authors mention that the magnetic orders and polar distortions of the 36 different cases has been verified by a symmetry analysis. What are the details of such a symmetry analysis? No reference? Like that it is difficult to verify what the authors checked through this symmetry analysis (I guess it is based on the loss of space inversion in the space group obtained from FINDYM, as supposed from caption of Table I?).

2- In line 167 to 174, the authors say that removing the 4f electrons of the Gd suppresses any onset of polarisation. This is right and this has been demonstrated from symmetry analysis before. Reference to these works would be welcome. For example in Phys. Rev. Lett. 108 219701 (2012), JETP Lett. 88 505 (2008), JPCM 28 123001 (2016), etc, etc.

3- the authors report the main phenomenological invariant terms that can be developed with the different order parameter present in these perovskite structures (magnetism on two sites, oxygen rotations, anti polar motions, shear strain). All along they say "it can be proved by symmetry that ...". However they never report the detailed symmetry analysis to allow the reader to check if they are correct or not. Here it would be very important to, at least, report the symmetry transformations of all the order parameters listed in the paper in a table (that can be added in the method section or a supplemental material) which will support the invariants they propose. This will also help a broader reader audience to understand the paper.

4- From line 657 to line 668, the authors discuss the case of ErCrO₃. To me this discussion is closed from the simple argument that by symmetry if the Er is not magnetised then the observed polarisation is not possible.

5- Line 669 to 700, same comment as for ErCrO₃. In SmFeO₃ why discussing so much details while it has been proved that by symmetry it is not possible to have such a polarisation state (Phys. Rev. Lett. 108 219701 (2012)). The results of the present paper also go toward this result. There is a controversy regarding these measurements and the authors could also mention that this polarisation might come from an extrinsic effect or improper measurement techniques (Phys. Rev. Lett. 113, 217203 (2014), JPCM 28 123001 (2016)).

6- In too many places in the manuscript the authors put sentences in brackets that are not well integrated in the text. I do not understand why because this makes the text way heavier to read. Should they be forwarded as footnotes (is it the format of Nature Communication for footnotes)? If not then these sentences should be integrated correctly in the main text with proper typography and connections with the other sentences. Here is the list of lines where these bracketed text should be acted:

- 199-202
- 237-238
- 265-270
- 300-303
- 317-321
- 343-345
- 350-353
- 400-402
- 458-463
- 564-567
- 601-606
- 651-656
- 763-765

In the method section:

7- The authors relaxed their structure up to 0.005 eV/Å. Is it sufficient to resolve polarisation as small as the one reported for RMO₃?

8- Why having performed the Berry phase calculations at the collinear level even for the structures obtained with the spin-orbit coupling (there is no technical reason to have done such a switch to compute the polarisation)? In view of the smallness of the polarisation such an approximation might be fatal. Doing so will give the same ionic contributions but the electronic part might be strongly affected, mostly in structures where the spin-orbit coupling is the key mechanism for ferroelectricity (though the authors focus on the collinear case afterward).

REVIEWERS' COMMENTS:

Reviewer #1 (Remarks to the Author):

The authors addressed my comments. I would suggest the publication of this work in Nature Comm.

However, I have one minor comment to be considered by the authors:

Since the authors now included a discussion on the mechanism of ferroelectricity in $\text{LaMn}_3\text{Cr}_4\text{O}_{12}$, the author might also discuss the recent theoretical finding in Phys. Rev. B 93, 174416 (2016) where the anisotropic symmetric exchange was proposed to explain the ferroelectricity.

Reviewer #2 (Remarks to the Author):

I checked the response to my previous comments. I'm satisfied by their response and revision. Thus, I'd like to recommend the acceptance.

Reviewer #3 (Remarks to the Author):

The authors went through all the referee's comments with care and details. Concerning my comments all of them have been addressed satisfactorily, mostly regarding the symmetries. The convergence of the calculations are quite clear and the text has been accordingly revised. I think the paper is ready for publication without any revision from my side and as mentioned in my first report I strongly encourage its publication in Nature Communication.

Reviewer #1 makes the following comments in his/her report:

“The authors studied the origin of magnetically-driven ferroelectric order in ABO₃ perovskite oxides with magnetic A and B cations that order in simple spin structures. They propose a magnetically-driven piezoelectric effect to explain the improper ferroelectricity. In my opinion, this is a rather new concept. Therefore, I would suggest the publication of this work in Nature Communications after the authors address the following comments:”

Answer: We appreciate very much this positive recommendation.

"(1) In Fig. 4, I would suggest the authors to add the (magnetic) point group of the three cases so that the reader can understand better the idea of magnetically-driven piezoelectric effect."

Answer: This is a very good point. Following the reviewer's suggestion, we have added the point group of the three cases in Fig. 4.

We should note here that, since in this example we are working with scalar spin variables, the macroscopic order parameters associated to the G-AFM order of the rare-earth and transition-metal sublattices do not have any spatial directionality, and we can take them to be invariant upon the action of all the atomic symmetry operations. Further, since time reversal symmetry is obviously broken in any magnetically-ordered configuration, it follows that the magnetic point groups coincide with the regular, atomic point groups.

"(2) In Phys. Rev. B 91, 100405(R) (2015), it was proposed that there exists an additional contribution to the electric polarization induced by the spin order in multiferroics due to the combined effects of magnetoelastic coupling and piezoelectric effect. In some sense, the magnetically-driven piezoelectric effect is closely related to the magnetically-driven ferroelectricity due to the piezoelectric effect. Thus, I would suggest the authors cite that work and discuss the similarity and difference between these related but different effects."

Answer: This is a good point. We have checked the suggested reference (Phys. Rev. B 91, 100405(R) (2015)). In this paper, the combined effect of magnetoelasticity and

piezoelectricity is shown to contribute to the electric polarization of magnetoelectric materials. That is to say, the change of magnetic ordering will induce strain via the magnetoelastic coupling, which in turn will result in a polarization change if the material is piezoelectric. This is also similar to the strain-mediated linear magnetoelectric response that occurs in materials that are simultaneously piezoelectric and piezomagnetic, which some of us demonstrated in Phys. Rev. Lett. 103, 267205 (2009).

In our case, the electric polarization of GdFeO₃ (GdCrO₃) also comes from the combination of magnetic ordering and piezoelectric effect. However, a critical difference is that in our case the material is not piezoelectric in absence of magnetic order; instead, the piezoelectricity is magnetically-driven. Further, in our case the strain is provided by the structural distortions that are typical of orthoferrites and orthochromites, and not by a magnetic order. So, while involving similar ingredients, the effect discussed in our paper is different from the one mentioned by the referee.

We have discussed these issues in our revised manuscript, and also cited the reference provided by the reviewer.

“(3) Typo: Line 314 "In order to indentify””

Answer: We have corrected this typo thanks to this reviewer.

Reviewer #2 had the following comments in his/her report:

“In this work, Zhao et al. performed very detailed DFT calculations on multiferroic RFeO₃ and RCrO₃. The underlying physics of magnetism-induced-polarization has been clearly illustrated. Their conclusion is that the exchange striction dominantly contributes to the polarization once the magnetic moments of R and Fe/Cr are collinear. The roles of octahedral rotations, anti-phase motion of R, and shear distortion have been clarified. In addition, phenomenological Landau energy items have been formulated. This work is well-done and will be physical interesting to the multiferroic community. I have following comments:”

Answer: We appreciate very much this positive assessment of our work.

“1. The title is too broad. The systems under study only involve $RFeO_3$ and $RCrO_3$. Although the mechanisms revealed in the present work can be extended to some similar systems, it remains too broad to cover all multiferroics with two magnetic sublattices. For example, the RMn_2O_5 , $CaMn_7O_{12}$, and $LaMn_3Cr_4O_{12}$ systems also own two magnetic sublattices, which own different stories.”

Answer: This is a good point. As a result, we change our title to “Improper electric polarization in simple perovskite oxides with two magnetic sublattices”, to make the distinction between simple perovskites (as the ones considered in this work) and structurally more complex (layered, chemically ordered) compounds as the ones mentioned by the reviewer.

Nevertheless, since the basic conclusions of the work are applicable to the whole family of simple perovskites with two magnetic sublattices, we would like to retain a rather general title, avoiding the notion that the conclusions of our work only apply to ferrites and chromites.

“2. Fig 4c is helpful to understand the exchange striction between Fe and R. However, according to Fig. 3a and 3b, even without the shear strain, considerable polar forces still exist, in the same order of magnitude with that in 3c. Then what's the physical mechanism for Fig. 3a and 3b?”

Answer: A rather natural question, indeed. Ultimately, all the exchange-strictive effects quantified in Fig. 3 are driven by the symmetry breaking occurring when various lattice instabilities are condensed in combination with the spin order of the R and M sublattices. Now, some of these symmetry breakings are easier to visualize than others. We thus decided to include in the main text a figure of the most clear, and in our mind also most interesting, effect, i.e., the magnetically-induced piezoelectric response.

Nevertheless, in the revised manuscript we have included a panel in Fig. 4 (panel d) showing how the anti-polar motion of the R cations induces a force on the M atoms (this is related with the interaction in Eq. (5)). The situation is a bit more complex in this case, but we believe the figure is still clear. A similar figure could be prepared for the anti-phase rotations of the oxygen octahedra, but it is considerably more crowded and difficult to visualize; hence, since such a figure does not provide any qualitatively new information, we have not included it in the paper.

“3. I agree with the exchange striction. But I'm suspected on the detailed microscopic origin. In lines 310-313, the authors argued that "they involve regular (super) exchange mechanisms". Is there any solid evidence to support the superexchange between Fe/Cr and R? I'm quite suspected on this point. The authors need to check the energy band structure. If their bands are far from each other in energy (very possible), the superexchange should be negligible. A hint to support my guess is that the polarization disappears if the lattice is fixed. Previous DFT studies on many type-II multiferroics with exchange striction revealed significant contribution from pure electronic part, which is a hint of superexchange involved. Considering this fact and the very small polarization (comparing with other exchange-strictive multiferroics), I guess the interaction between Fe/Cr and R is mainly the magnetic dipole-dipole interaction, instead of the real superexchange.”

Answer: We have to thank the reviewer for this very good point. When we wrote “*regular (super) exchange mechanisms*” we just wanted to emphasize that the couplings responsible for the effects observed occur in absence of spin-orbit interactions, and are thus non-relativistic. We have modified the text in the manuscript to correct this issue.

Obviously, it was not our goal in this paper to model the magnetic interactions between spin sublattices in detail. Nevertheless, let us note that the plots below, showing the computed density of states for GdFeO₃ and GdCrO₃, indicate a significant overlap of the Gd-4f and the 3d bands of the transition metals, particularly in the case of the ferrite. Hence, we guess that super-exchange might have a contribution to the R-M interactions in the studied compounds, but a detailed analysis, and construction of a spin Hamiltonian, remains for future work.

“4. I noticed that the dual G-type collinear antiferromagnetism in recently found LaMn3Cr4O12 also contributed to a polarization. However, the underlying physical mechanism, involving the spin-orbit coupling, is quite different from the current one. The authors need to clarify this point.”

Answer: We thank the reviewer for directing our attention to this interesting compound. We have checked the article on LaMn3Cr4O12 by Wang et al., which has recently appeared in Phys. Rev. Lett. 115, 087601 (2015). According to this work, the magnetic ordering for Cr sublattice is indeed G-type. However, the magnetic order for Mn sublattice is not exactly G-type (please note that in the cubic cell, there are 8 Cr ions, but only 6 Mn ions). As a result, the dual G-type collinear antiferromagnetism for LaMn3Cr4O12 differs significantly from that in RMO3 perovskites.

In addition, we have examined the experimentally reported crystal structure of LaMn3Cr4O12 (Long et al, J. Am. Chem. Soc. 131, 16244–16247 (2009)). This has allowed us to determine the symmetry of such oxide considering scalar magnetism and three-dimensional magnetism. We find that the scalar (dual G-type) magnetism results in a non-polar point group, at variance with what happens in the orthoferrites and orthochromites. In contrast, the three-dimensional magnetism (dual G-type for Cr and Mn along [111] or [-1-1-1] directions) renders a polar point group, even if the Cr and Mn spins remain collinear. We agree that, in such a case, the occurrence of magnetically-driven polar distortions relies on a spin-orbit mechanism. Hence, LaMn3Cr4O12 differs from our RMO3 compounds in that aspect as well.

In the revised version of our manuscript we cite the articles on $\text{LaMn}_3\text{Cr}_4\text{O}_{12}$, briefly explaining the differences with our current results and conclusions.

“5. The discussion on SmFeO_3 seems to be needless to me. The experimental result on SmFeO_3 are quite suspicious and negatively commented by following works. The physics proposed here can not say anything on SmFeO_3 . So I suggest the authors to discard the related discussion.”

Answer: We thank the reviewer for pointing out this issue about SmFeO_3 . We were aware of this, but thought that SmFeO_3 still deserved a brief note in our paper. Seeing this recommendation, which aligns with point 5 of the third reviewer, we have drastically shortened this discussion. Yet, we have kept a brief note, in connection with the similar case of ErCrO_3 , for the benefit of non-expert readers who may run into (some of) the SmFeO_3 literature and find it difficult to reconcile it with the conclusions of our work.

Reviewer #3 had the following comments in his/her report:

“In this manuscript the authors scrutinise from state of the art DFT calculations the microscopic origin of the electric polarisation observed in rare-earth orthoferrites and orthochromites. They first consider all the possible non-collinear magnetic configurations of both rare-earth and transition metal cations in GdFeO₃ and compute the resulting polarisation. They found that 13 cases over 36 are indeed polar. Over the 13 cases they distinguish that two cases can be separated: Those requiring spin-orbit coupling and those requiring only exchange-striction effect. They observe that the later has the largest induced polarisation and it is the keystone of the present paper. Then, they prove that this mechanism is present when both rare-earth and transition-metal atoms adopt parallel G-type AFM vector and is independent of the presence of a weak ferromagnetic component. After having analysed the magnetic origin of the induced polarisation the authors explore what are the structural key parameters that cooperate with the magnetism to induce the polarisation. Here they originally proved that the presence of antiphase rotation of the octahedra and a shear strain are mandatory to observe the induced polarisation. They then show from a phenomenological-like model how this magnetostructural order can appear in these perovskite structures to couple with the polarisation. The paper is well written and, beside a few comments I’ll give below, the results are sufficiently striking, novel and important to be accepted for publication in Nature Communication. Additionally the paper will represent a reference and high impact article in view of the present increase of interest for rare-earth based perovskites.”

Answer: We appreciate the excellent summary of our work made by the reviewer, and his/her positive recommendation.

“1- From line 150 to 155, the authors mention that the magnetic orders and polar distortions of the 36 different cases has been verified by a symmetry analysis. What are the details of such a symmetry analysis? No reference? Like that it is difficult to verify what the authors checked through this symmetry analysis (I guess it is based on the loss of space inversion in the space group obtained from FINDYM, as supposed from caption of Table 1?).”

Answer: We obtain the magnetic symmetries with the FINDSYM code, as mentioned in the Method part. The FINDSYM code allows the user to get the crystal space group as well as the magnetic space group (by selecting the option of “Include magnetic

moments. Give components along the three vectors above which define a conventional unit cell.”) The details about FINDSYM can be found at “<http://stokes.byu.edu/iso/findsym.php>”, which is now cited in the revised Methods section.

In the revised manuscript, we have explicitly indicated in the Methods section that FINDSYM can be used to tackle magnetic symmetries.

“2- In line 167 to 174, the authors say that removing the 4f electrons of the Gd suppresses any onset of polarisation. This is right and this has been demonstrated from symmetry analysis before. Reference to these works would be welcome. For example in Phys. Rev. Lett. 108 219701 (2012), JETP Lett. 88 505 (2008), JPCM 28 123001 (2016), etc, etc.”

Answer: We apologize for this omission. We cite the articles mentioned by the reviewer in the revised manuscript.

“3- the authors report the main phenomenological invariant terms that can be developed with the different order parameter present in these perovskite structures (magnetism on two sites, oxygen rotations, anti polar motions, shear strain). All along they say “it can be proved by symmetry that ...”. However they never report the detailed symmetry analysis to allow the reader to check if they are correct or not. Here it would be very important to, at least, report the symmetry transformations of all the order parameters listed in the paper in a table (that can be added in the method section or a supplemental material) which will support the invariants they propose. This will also help a broader reader audience to understand the paper.”

Answer: We agree with the reviewer in that it is always a good idea to popularize the use of symmetry theory. Following his/her advice, in the Methods section of the revised manuscript, we have included a table showing how the considered order parameters transform under the generators of the full cubic point group of the ideal perovskite structure, to facilitate verification of the invariants discussed in our text.

“4- From line 657 to line 668, the authors discuss the case of ErCrO₃. To me this discussion is closed from the simple argument that by symmetry if the Er is not magnetised then the observed polarisation is not possible.”

Answer: We thank the reviewer for this comment. While we agree with his/her opinion, we think it is a good idea to briefly comment on ErCrO₃ in the text, in the interest of non-expert readers for whom this point may not be so trivial.

“5- Line 669 to 700, same comment as for ErCrO₃. In SmFeO₃ why discussing so much details while it has been proved that by symmetry it is not possible to have such a polarisation state (Phys. Rev. Lett. 108 219701 (2012)). The results of the present paper also go toward this result. There is a controversy regarding these measurements and the authors could also mention that this polarisation might come from an extrinsic effect or improper measurement techniques (Phys. Rev. Lett. 113, 217203 (2014), JPCM 28 123001 (2016)).”

Answer: We again agree. Following the reviewer’s advice (which coincides with the point 5 made by the second reviewer) we have drastically shorten our discussion on SmFeO₃, but still mention it, together with ErCrO₃, for the benefit of non-expert readers.

“6- In too many places in the manuscript the authors put sentences in brackets that are not well integrated in the text. I do not understand why because this makes the text way heavier to read. Should they be forwarded as footnotes (is it the format of Nature Communication for footnotes?)? If not then these sentences should be integrated correctly in the main text with proper typography and connections with the other sentences. Here is the list of lines where these bracketed text should be acted:- 199-202- 237-238- 265-270- 300-303- 317-321- 343-345- 350-353- 400-402- 458-463- 564-567- 601-606- 651-656- 763-765”

Answer: We have revised the text to make it more readable. More specifically, we have removed the brackets in all but three of the instances mentioned by the reviewer.

“7- The authors relaxed their structure up to 0.005 eV/Angstrom. Is it sufficient to resolve polarisation as small as the one reported for RMO3?”

Answer: In order to address this issue, we optimized the crystal structures for GdFeO3 and GdCrO3 down to 0.001 eV/Ang. level. When doing the structural optimization, we considered four typical cases for each oxide, namely, the scalar magnetic case and the non-collinear magnetic structures labeled as Ga/Ga, Gb/Gb and Gc/Gc in our paper. Using the optimized crystal structures, we then computed their polarization at the scalar-magnetism level; the results are shown in Table I below, clearly proving that the force convergence criteria of 0.005 eV/Ang. is sufficient to compute accurately the magnetically-induced polarizations.

This convergence check is now briefly mentioned in the Methods section of the revised manuscript.

Table I: Computed polarization P_c of GdFeO₃ and GdCrO₃ in the Gd/Fe spin arrangements. The unit of polarization is $\mu\text{C}/\text{cm}^2$.

	G _a /G _a	G _b /G _b	G _c /G _c	Scalar
GdFeO3 (<0.005 eV/Ang.)	-0.42	-0.41	-0.41	-0.40
GdFeO3 (<0.001 eV/Ang.)	-0.42	-0.42	-0.42	-0.41
GdCrO3 (<0.005 eV/Ang.)	-0.14	-0.15	-0.14	-0.16
GdCrO3 (<0.001 eV/Ang.)	-0.15	-0.15	-0.15	-0.15

“8- Why having performed the Berry phase calculations at the collinear level even for the structures obtained with the spin-orbit coupling (there is no technical reason to have done such a switch to compute the polarisation)? In view of the smallness of the polarisation such an approximation might be fatal. Doing so will give the same ionic contributions but the electronic part might be strongly affected, mostly in structures where the spin-orbit coupling is the key mechanism for ferroelectricity (though the authors focus on the collinear case afterward).”

Answer: This is a valid concern. To check this we have computed the polarization of GdFeO₃ and GdCrO₃ considering the spin-orbital coupling (soc). The results, shown in Table II below, are essentially identical to those obtained when the polarization is computed without soc, but using the structure relaxed in a calculation including soc.

Additionally, let us note that Stroppa et al. [New Journal of Physics 12, 093026 (2010); see page 8, lines 16 and 21] report the total polarization of DyFeO₃ to be $\sim 0.20 \mu\text{C}/\text{cm}^2$ when computed at the scalar-magnetism level without soc, while it turns to be $0.18 \mu\text{C}/\text{cm}^2$ when including the soc. This further suggests, in agreement with our observations, that the effect of spin-orbit coupling on the electronic part of the polarization is rather small in compounds of this family.

This check is now briefly mentioned in the Methods section of the revised manuscript.

Table II: Computed polarization P_c of GdFeO₃ and GdCrO₃ in the Gd/Fe spin arrangements. The unit of polarization is $\mu\text{C}/\text{cm}^2$.

	G_a/G_a	G_b/G_b	G_c/G_c
GdFeO ₃ (without soc)	-0.42	-0.41	-0.41
GdFeO ₃ (withsoc)	-0.42	-0.41	-0.41
GdCrO ₃ (without soc)	-0.14	-0.15	-0.14
GdCrO ₃ (with soc)	-0.14	-0.15	-0.14

Summary of changes

Our changes, which are summarized below, are also marked by red in our revised manuscript.

- In response to the comment #1 from Reviewer #1:

We have modified Fig. 4 to include the point groups of the different structures.

- In response to the comment #2 from Reviewer #1:

We have inserted the following paragraph in the discussion section:

“Interestingly, the magnetically-driven piezoelectric response, and ensuing improper polarization, revealed in this work can be viewed as an example of a broader family of effects. Thus, for instance, it has been recently shown that spin order may result in a contribution to the polarization of a ferroelectric via a combination of magnetoelastic and piezoelectric couplings[28]. Similarly, it is possible to obtain a linear magnetoelectric response in materials that are both piezomagnetic and piezoelectric[29]. Nevertheless, while they involve similar ingredients, there is a critical difference between such effects and the one that we have introduced in this work: in the present case the material is not piezoelectric in absence of magnetic order; instead, piezoelectricity is magnetically driven. Hence, we could say that the mechanisms discussed in Refs.28 and 29 are typical of the so-called “type I” multiferroics[17] -- where ferroelectricity is a *proper* order --, while the mechanisms discussed here would generally correspond to “type II” materials with an improper (magnetically-induced) polar order.”

We also added the corresponding new literature citations.

- In response to the comment #3 from Reviewer #1:

We have corrected the indicated typo.

- In response to the comment #1 from Reviewer #2:

We have changed the title to “Improper electric polarization in simple perovskite oxides with two magnetic sublattices”.

- In response to the comment #2 from Reviewer #2:

We have added a new panel to Fig. 4, to illustrate the exchange-strictive mechanism associated to anti-polar motions of the rare-earth cations. We have also modified panel (c) of Fig. 4 to better show how the combined action of spin order and anti-polar distortions creates equal polar forces throughout the lattice.

- In response to the comment #3 from Reviewer #2:

We have modified the problematic sentence. The new version reads:

“Note that such couplings will be of the so-called exchange-strictive type, as our simulations indicate that they occur even in absence of relativistic spin-orbit interactions.”

- In response to the comment #4 from Reviewer #2, we added sentences that read:

“Our present results are somewhat reminiscent of recent observations on materials like $\text{LaMn}_3\text{Cr}_4\text{O}_{12}$ [30,31], which presents a “dual G-type collinear antiferromagnetism” that results in an electric polarization [30]. However, beyond obvious structural and stoichiometric differences, it should be noted that in $\text{LaMn}_3\text{Cr}_4\text{O}_{12}$ the magnetically-driven polar distortion relies on the spin-orbit couplings, at variance with the main effects discussed here.”

- In response to the comment #5 from Reviewer #2 and comments #4 and #5 from Reviewer #3:

We have removed the detailed discussions on ErCrO₃ and SmFeO₃, and inserted this brief paragraph instead:

“Also, let us note that various works on ferrites (like e.g. SmFeO₃ [12]) and chromites (like e.g. ErCrO₃ [13]) have reported electric polarizations occurring at temperatures well above the ordering transition for the rare-earth spins. However, according to our present results and to the symmetry analysis made by others [18-20], a simple AFM order of the Fe/Cr spins cannot produce such polarizations in absence of rare-earth magnetism; instead, a cycloidal arrangement -- as e.g. the one occurring in prototypic compound TbMnO₃ (Ref. 32) – is needed for a polarization to appear.”

- In response to the comment #1 from Reviewer #3:

We have revised the corresponding sentence in the Methods section, which now reads:

“The space groups of various configurations are identified using the FINDSYM code, which is able to tackle magnetic symmetries [37,38].”

- In response to the comment #2 from Reviewer #3:

We have modified a relevant sentence, which now reads:

“This further demonstrates that the existence of a magnetic ordering at both the rare-earth and Fe/Cr sublattices is necessary to obtain an electric polarization, and is in agreement with the symmetry analysis made by various authors [18-20].”

- In response to the comment #3 from Reviewer #3:

We have inserted the requested table in the Methods section.

- See above for changes in response to comments #4 and #5 from Reviewer #3.

- In response to the comment #6 from Reviewer #3:

We have revised the text in all but three of the instances mentioned by the Reviewer.

- In response to the comment #7 from Reviewer #3:

In the Methods section, we have revised a couple of sentences, which now read:

“The structural optimizations are carried out until the forces acting on the atoms go below $0.005 \text{ eV}/\text{\AA}$; we checked this is sufficient by running relaxations down to $0.001 \text{ eV}/\text{\AA}$ in a few representative cases.”

and

“We typically run the Berry phase calculations at the scalar-magnetism level, but using the structures obtained from non-collinear calculations including spin-orbit couplings; the polarizations thus computed were checked to be correct in a few representative cases for which we ran the Berry phase calculation at the non-collinear level with spin-orbit.”

Additionally, we have corrected panel (c) of Fig. 3, we found we had made a factor-of-two mistake in the calculation of the shear strain. For the benefit of interested readers, we now indicate how we compute the shear strains in the Methods section.

REVIEWERS' COMMENTS:

Reviewer #1 (Remarks to the Author):

The authors addressed my comments. I would suggest the publication of this work in Nature Comm.

However, I have one minor comment to be considered by the authors: Since the authors now included a discussion on the mechanism of ferroelectricity in $\text{LaMn}_3\text{Cr}_4\text{O}_{12}$, the author might also discuss the recent theoretical finding in Phys. Rev. B 93, 174416 (2016) where the anisotropic symmetric exchange was proposed to explain the ferroelectricity.

Answer: We thank this reviewer very much for his/her recommendation for the publication of our manuscript. Following the his/her suggestion, in the revised paper we cite the reference mentioned.

Reviewer #2 (Remarks to the Author):

I checked the response to my previous comments. I'm satisfied by their response and revision. Thus, I'd like to recommend the acceptance.

Answer: We thank this reviewer very much for his/her recommendation for publication of our manuscript.

Reviewer #3 (Remarks to the Author):

The authors went through all the referee's comments with care and details. Concerning my comments all of them have been addressed satisfactorily, mostly regarding the symmetries. The convergence of the calculations are quite clear and the text has been accordingly revised. I think the paper is ready for publication without any revision from my side and as mentioned in my first report I strongly encourage its publication in Nature Communication.

Answer: We thank this reviewer very much for his/her recommendation for publication of our manuscript.